# Oral Lactoferrin Supplementation during Induction Chemotherapy Promotes Gut Microbiome Eubiosis in Pediatric Patients with Hematologic Malignancies

**DOI:** 10.3390/pharmaceutics14081705

**Published:** 2022-08-16

**Authors:** Federica D’Amico, Nunzia Decembrino, Edoardo Muratore, Silvia Turroni, Paola Muggeo, Rosamaria Mura, Katia Perruccio, Virginia Vitale, Marco Zecca, Arcangelo Prete, Francesco Venturelli, Davide Leardini, Patrizia Brigidi, Riccardo Masetti, Simone Cesaro, Daniele Zama

**Affiliations:** 1Microbiomics Unit, Department of Medical and Surgical Sciences (DIMEC), University of Bologna, 40138 Bologna, Italy; 2Neonatal Intensive Care Unit-AOU Policlinico “Rodolico-San Marco”, University of Catania, 95131 Catania, Italy; 3Pediatric Hematology/Oncology, Fondazione IRCCS Policlinico San Matteo, 27100 Pavia, Italy; 4Pediatric Hematology and Oncology Department, IRCCS Azienda Ospedaliero-Universitaria di Bologna, 40138 Bologna, Italy; 5Unit of Microbiome Science and Biotechnology, Department of Pharmacy and Biotechnology, University of Bologna, 40126 Bologna, Italy; 6Pediatric Hematology and Oncology Department, University of Bari, 70121 Bari, Italy; 7Pediatric Hematology and Oncology Department, “A Cao” Microcitemic Pediatric Hospital, “Botzu” Medical Center, 09100 Cagliari, Italy; 8Pediatric Hematology and Oncology Department, “Santa Maria della Misericordia” Hospital, 06132 Perugia, Italy; 9Pediatric Hematology and Oncology, Department of Mother and Child, Azienda Ospedaliera Universitaria Integrata, 37126 Verona, Italy; 10Department of Experimental, Diagnostic and Specialty Medicine (DIMES), University of Bologna, 40138 Bologna, Italy; 11Department of Medical and Surgical Sciences (DIMEC), University of Bologna, 40138 Bologna, Italy; 12Pediatric Emergency Unit, IRCCS Azienda Ospedaliero-Universitaria di Bologna, 40138 Bologna, Italy

**Keywords:** gut microbiota, lactoferrin, chemotherapy, hematologic malignancies, oral supplementation, eubiosis, pediatrics, acute lymphoblastic leukemia

## Abstract

Induction chemotherapy is the first-line treatment for pediatric patients with hematologic malignancies. However, several complications may arise, mainly infections and febrile neutropenia, with a strong impact on patient morbidity and mortality. Such complications have been shown to be closely related to alterations of the gut microbiome (GM), making the design of strategies to foster its eubiosis of utmost clinical importance. Here, we evaluated the impact of oral supplementation of lactoferrin (LF), a glycoprotein endowed with anti-inflammatory, immunomodulatory and antimicrobial activities, on GM dynamics in pediatric oncohematologic patients during induction chemotherapy. Specifically, we conducted a double blind, placebo-controlled trial in which GM was profiled through 16S rRNA gene sequencing before and after two weeks of oral supplementation with LF or placebo. LF was safely administered with no adverse effects and promoted GM homeostasis by favoring the maintenance of diversity and preventing the bloom of pathobionts (e.g., *Enterococcus*). LF could, therefore, be a promising adjunct to current therapeutic strategies in these fragile individuals to reduce the risk of GM-related complications.

## 1. Introduction

Hematologic malignancies, including leukemias and lymphomas, are the most frequent types of cancer in pediatric patients [1]. Current long-term survival rates can reach 90% thanks to chemotherapy and/or immunotherapies, but several complications may arise, particularly infections, with a strong impact on the patient’s morbidity and mortality [2].

The gut microbiome (GM)—the set of all microorganisms colonizing the gastrointestinal tract—has been shown to have a strategic role not only during the physiological development of the child [3], but also in the pathological context, including the oncological setting [4,5]. Indeed, GM alterations have been found in several types of cancer, including hematologic malignancies, and are suggested to trigger their development [6,7,8], and/or contribute to their progression, also influencing the efficacy of therapeutic interventions [8,9,10,11,12]. On the other hand, it is known that chemo-immunotherapy treatments can drastically impact GM, favoring the establishment of unbalanced low-diversity profiles, sometimes monodominated by opportunistic pathogens (e.g., *Enterococcus*) [13,14,15], with potentially severe health implications even in the long-term. Indeed, distinct dysbiotic features and dynamics during chemo-immunotherapy have been associated with an increased rate of complications, including the onset of bloodstream infections (BSIs) and febrile neutropenia, increased relapses and reduced survival [14,16,17]. It is, therefore, highly plausible to hypothesize that promoting the maintenance of GM homeostasis during anticancer therapies could be instrumental in preventing the development of serious complications, and therefore reducing treatment-related mortality.

In recent years, various strategies have been used for this purpose, mainly based on antibiotic sparing guidelines, nutritional support and prebiotic/probiotic supplementation [18,19,20,21]. Lactoferrin (LF), a bioactive glycoprotein of bovine and human milk, has shown anti-inflammatory, immunomodulatory and antimicrobial activities relevant to the maintenance of GM-host mutualism [22,23]. Several clinical studies have been conducted on the usefulness of its supplementation in children, particularly in preterm neonates, in whom bovine LF administration was associated with reduced incidence of late-onset sepsis and necrotizing enterocolitis [24,25,26]. Indeed, LF has been shown to strengthen the intestinal epithelial barrier, promote antagonism towards enteropathogens and provide signals to balance anti- and pro-inflammatory responses [27,28,29]. Furthermore, no adverse effects secondary to its supplementation have been reported [24,25,26], suggesting that LF can be safely administered to other patient groups even at critical times, such as cancer patients during chemotherapy.

Here, we explored, for the first time, the potential of LF in counteracting chemotherapy-related GM dysbiosis. Specifically, we enrolled pediatric patients at the onset of hematologic malignancies and profiled their GM, through next-generation 16S rRNA gene sequencing, before and after two weeks of oral supplementation with LF or placebo during induction chemotherapy. According to our results, supplementation with LF promotes the maintenance of GM diversity and prevents the overabundance of pathobionts, thus representing a promising adjunct to current intervention strategies in these fragile individuals.

## 2. Methods

### 2.1. Study Design and Patient Enrolment

The present work represents the associated sub-study on GM of a randomized, double-blind, placebo-controlled trial of oral LF supplementation in pediatric cancer patients at high risk of febrile neutropenia. The study was proposed by the Supportive Therapy Working Group of Italian Pediatric Hematology and Oncology Association (AIEOP) and approved by the ethics committees of the Fondazione IRCCS Policlinico San Matteo, Pavia, Italy (Protocol number 20160001176), and ratified by each participating institution. Parents or guardians provided written informed consent, according to the declaration of Helsinki. International clinical trial registration was not provided considering that in the present paper we report the results of a sub-study on gut microbiome of a larger randomized clinical trial focused on clinical outcomes. Between 1 January 2017, and 30 June 2020, eligible pediatric patients older than 1 month undergoing first-line induction chemotherapy for acute lymphoblastic leukemia (ALL), acute myeloid leukemia (AML) and non-Hodgkin’s lymphoma (nHL) from nine Italian pediatric oncohematologic centers (Fondazione IRCCS Policlinico San Matteo of Pavia, IRCCS Azienda Ospedaliero-Universitaria of Bologna, University Hospital of Verona, “Santa Maria della Misericordia” Hospital of Perugia, University of Bari, “A Cao” Microcitemic Pediatric Hospital “Botzu” Medical Center of Cagliari, Casa Sollievo della Speranza Hospital San Giovanni Rotondo, Fondazione Monza e Brianza per il Bambino e la Mamma, IRCCS Burlo Garofalo Trieste) were prospectively enrolled. Patients were randomized into two groups: one received bovine LF (Mosiac^®^, Pharmaguida, Rome, Italy) at a dose of 200 mg/die once daily for two months, while the other received placebo. The complete composition of both oral bovine LF and placebo is available in Appendix A. Supplementation was initiated from the start of induction chemotherapy. Children were allocated to the two groups in a 1:1 ratio. Randomization was stratified by center, with blocks of different size (2-4-6). The random allocation sequence was generated using the software STATA 14.0 (StataCorp., College Station, TX, USA). Whenever a patient was enrolled in the study, the informatics system communicated to the clinician the patient’s allocation to group A or B. Both LF and placebo were supplied by the same company in the same pharmaceutical form. The company labelled each pack of medication as A or B based on the computer-generated randomization list. Clinical and research staff remained blinded to study group assignments throughout the study. Exclusion criteria included gut colonization by multidrug-resistant bacteria before cancer diagnosis. During the study period, every center provided supportive therapy according to local guidelines, including antibiotic prophylaxis. Antibiotics were recorded in the electronic CRF.

For this sub-study on GM, only patients with at least two fecal samples available were included, i.e., one prior to the start of LF or placebo supplementation and the other after 14 days. Fecal samples were stored at −80 °C and shipped in dry ice to the Department of Pharmacy and Biotechnology (University of Bologna, Bologna, Italy) for GM analysis.

### 2.2. Microbial DNA Extraction and 16S rRNA Gene Sequencing

Microbial nucleic acids were extracted from 0.25 g of each stool sample using the repeated bead-beating plus column method, as previously described with few modifications [19,30,31]. In brief, all samples were added with 1 mL of lysis buffer (500 mM NaCl, 50 mM Tris-HCl pH 8, 50 mM EDTA, and 4% SDS) and bead-beaten three times in the presence of four 3-mm glass beads and 0.5 g of 0.1 mm zirconia beads (BioSpec Products, Bartlesville, OK, USA) using a FastPrep instrument (MP Biomedicals, Irvine, CA, USA), set to 5.5 movements/s for 1 min. After an incubation step at 95 °C for 15 min, samples were centrifuged at 13,000 rpm to separate cell debris and incubated with 260 μL of 10 M ammonium acetate. Supernatants were recovered by centrifugation at 13,000 rpm and incubated in ice for 30 min with one volume of isopropanol. Precipitated nucleic acids were washed with 70% ethanol, re-suspended in 100 μL of TE buffer (10 mM Tris-HCl, 1 mM EDTA pH 8.0), and incubated at 37 °C for 15 min with 2 μL of 10 mg/mL DNase-free RNase. DNA purification was performed using the DNeasy Blood and Tissue Kit (QIAGEN, Hilden, Germany) following the manufacturer’s instructions. DNA concentration and quality were evaluated using the NanoDrop ND-1000 spectrophotometer (NanoDrop Technologies, Wilmington, DE, USA).

The V3–V4 hypervariable regions of the 16S rRNA gene were amplified using the 341F and 785R primers with linked Illumina adapter overhang sequences [32]. Fragment amplification was performed using the KAPA HiFi Hot Start Ready Mix (Roche, Basel, Switzerland), and the thermocycle was set up following the Illumina protocol “16S Metagenomic Sequencing Library Preparation” (Illumina, San Diego, CA, USA). Libraries were purified with a magnetic bead-based clean-up system (Agencourt AMPure XP; Beckman Coulter, Brea, CA, USA), and a limited-cycle PCR was performed using Nextera technology to obtain the indexed libraries, followed by a second clean-up step. Final libraries were prepared by pooling all samples at an equimolar concentration of 4 nM. The pool was then denatured and diluted to 5 pM before sequencing on an Illumina MiSeq platform with a 2 × 250 bp paired-end protocol according to manufacturer’s instructions.

### 2.3. Bioinformatics and Statistics

Raw sequences were processed using a pipeline combining PANDASeq [33] and QIIME 2 [34]. After length and quality filtering, reads were binned into amplicon sequence variants (ASVs) using the DADA2 pipeline [35]. Taxonomic assignment was carried out using the VSEARCH algorithm [36] and the Greengenes database. Chimeras were discarded during the analysis workflow. GM sequences of healthy subjects matched to cancer patients for major GM-associated confounding factors (i.e., age, sex, and body mass index—BMI) [37] were used as controls. Sequences were from different cohorts to minimize study-related bias, specifically from: Biagi et al., 2016 (deposited in MG-RAST database: project ID mgp17761) [38], Rampelli et al., 2018 (MG-RAST database: from mgm4780879.3 to mgm4781018.3) [39] and Muleviciene et al., 2018 [40]. It should be mentioned that the control samples had been processed in the same laboratory as the study samples for DNA extraction, library preparation, and sequencing, thereby further reducing any potential experimental bias. Alpha diversity was evaluated using different metrics: inverse Simpson index, Faith’s PD (phylogenetic diversity) index and number of observed ASVs. Bray–Curtis dissimilarities were used to construct Principal Coordinates Analysis (PCoA) graphs.

All statistical analyses were performed using R software. PCoA plots were generated using the “vegan” and “Made4” [41] packages, and data separation was tested by permutation test with pseudo-F ratio (function “Adonis” in “vegan”). Hierarchical Ward-linkage clustering based on Pearson’s correlation coefficients of the relative abundance of genera was performed with the “Made4” package. The Wilcoxon test, paired or unpaired as needed, was used to compare differences in alpha diversity and GM composition between groups. *p* values were corrected for multiple comparisons using the Benjamini–Hochberg method. A false discovery rate (FDR) ≤ 0.05 was considered as statistically significant while FDR between 0.05 and 0.1 as a trend.

Qualitative clinical variables were compared using Fisher’s exact test while quantitative ones were compared using the Mann–Whitney U test.

## 3. Results

### 3.1. Study Cohort Description

Thirty-four pediatric patients affected by hematologic malignancies were enrolled by six of the nine Italian centers participating in the main study (Fondazione IRCCS Policlinico San Matteo of Pavia, IRCCS Azienda Ospedaliero-Universitaria of Bologna, University Hospital of Verona, “Santa Maria della Misericordia” Hospital of Perugia, University of Bari, “A Cao” Microcitemic Pediatric Hospital “Botzu” Medical Center of Cagliari). Patient characteristics are summarized in Table 1. Patients were divided into two groups: (i) those who received 200 mg/die LF supplementation, and (ii) those who received placebo. The two groups were homogeneous in age, diagnosis and distribution among centers. Weight before initiation of therapy was lower in patients receiving LF rather than placebo (*p* = 0.03). No differences were observed between groups in antibiotic use in the 30 days prior to therapy and during induction chemotherapy.

Interestingly, febrile neutropenia occurred less frequently in patients receiving LF (57.1% vs. 90% in the placebo group; *p* = 0.04). No adverse effects were reported in any patient.

For each patient, fecal sampling was performed before starting LF or placebo supplementation and chemotherapy and after 2 weeks. A total of 68 fecal samples were analyzed through 16S rRNA gene sequencing, yielding 2,993,592 high-quality reads (mean ± SD; 44,023 ± 7449).

### 3.2. The GM Profile of Pediatric Oncohematologic Patients Segregates from That of Healthy Controls

The GM profile of patients at baseline (i.e., before starting chemotherapy and supplementing with LF or placebo) was compared with that of as many healthy subjects from previous studies [38,39,40], matched by major GM-associated confounding variables (i.e., age, sex, and BMI) [37]. Patients showed significantly less alpha diversity, as estimated by the inverse Simpson index (Wilcoxon test, *p* = 0.008) (Figure 1A). Moreover, PCoA based on Bray–Curtis dissimilarities highlighted a significant separation in GM structure between patients and controls (permutation test with pseudo-F ratio, *p* = 0.001) (Figure 1B). Regarding the patient cohort, no significant differences were observed by age group, prior antibiotic use, enrollment center and disease diagnosis, thus excluding any bias related to these variables (*p* ≥ 0.2) (Appendix A). From the taxonomic point of view, the GM profile of both patients and controls was dominated by the phylum Firmicutes (mean relative abundance in patients vs. controls, 61.8% vs. 59.4%) along with Actinobacteria (12.1% vs. 11.4%), and Proteobacteria (8.1% vs. 2.1%). However, patients showed reduced proportions of Bacteroidetes (Wilcoxon test, *p* < 0.001) (Appendix A). At the family level, the GM of both groups was mainly characterized by *Ruminococcaceae* (12.7% vs. 21.7%), *Lachnospiraceae* (14.0% vs. 21.6%), *Bacteroidaceae* (8.8% vs. 15.1%) and *Bifidobacteriaceae* (5.2 vs. 8.2%). However, all these families, along with *Prevotellaceae* and *Clostridiaceae*, were significantly underrepresented in patients (*p* ≤ 0.003) (Appendix A). As shown in the heatmap of Figure 1C, patients were depleted of typically health-associated genera from the *Lachnospiraceae* and *Ruminococcaceae* families, such as *Lachnospira*, *Dorea*, *Blautia*, *Coprococcus*, and *Ruminococcus*, as well as *Bacteroides* (*p* ≤ 0.02). On the other hand, they tended to show increased levels of pathobionts, namely [*Eubacterium*], *Coriobacteriaceae* taxa (i.e., *Adlercreutzia* and *Eggerthella*) and unclassified members of *Enterobacteriaceae* (*p* ≤ 0.1).

### 3.3. LF Supplementation during Induction Chemotherapy Counteracts GM Dysbiosis in Pediatric Oncohematologic Patients

Next, we assessed GM changes during induction chemotherapy, in relation to either LF or placebo supplementation. According to Faith’s PD index and the number of observed ASVs, alpha diversity decreased in patients receiving placebo (*p* ≤ 0.05), while it remained essentially unchanged in those who received LF (Figure 2A). PCoA based on Bray–Curtis distances showed significant segregation among groups (permutation test with pseudo-F ratio, *p* = 0.005) (Figure 2B). Moreover, a significant segregation of GM profiles by subject was found (*p* = 0.003) (Appendix A), emphasizing a high inter-individual variability. From the taxonomic standpoint, no significant differences were observed in the baseline GM layout between patients receiving LF and placebo (Wilcoxon test, *p* > 0.05). As for changes over time, increased proportions of the *Streptococcaceae* and *Veillonellaceae* families, and of the respective genera *Streptococcus* and *Veillonella*, were found in patients receiving LF, along with reduced proportions of *Bacteroides* (*p* ≤ 0.05). Furthermore, decreased levels of *Verrucomicrobiaceae* and *Ruminococcaceae* members were observed in the GM profile of all patients, regardless of supplementation (*p* ≤ 0.05). However, it should be noted that *Akkermansia*, the main genus of *Verrucomicrobiaceae*, was significantly reduced only in patients receiving LF, while *Oscillospira*, a health-associated taxon belonging to *Ruminococcaceae*, was reduced only in those receiving placebo (*p* ≤ 0.05). On the other hand, patients who received placebo showed an increase over time in *Enterobacteriaceae*, *Lactobacillaceae* and *Enterococcaceae* (particularly *Enterococcus*), as well as a decrease in *Rikenellaceae* (*p* ≤ 0.05) (Figure 3).

Given the differences in baseline weight between the study groups, analyses were repeated only in those patients weighing less than 20 kg (and younger than 6 years of age) (Appendix A). Even considering this subgroup, patients receiving placebo showed a temporal decrease in alpha diversity (for both the number of observed ASVs and Faith’s PD metrics, Wilcoxon test, *p* < 0.05), and significantly segregated from those receiving LF in the Bray–Curtis-based PCoA (permutation test with pseudo-F ratio, *p* = 0.03). Moreover, the genus-level signatures identified above were overall confirmed, although only a trend was found for some taxa due to the smaller number of patients (Wilcoxon test, *p* ≤ 0.1).

## 4. Discussion

As far as we know, for the first time here, we provided insights into the impact of LF supplementation during induction chemotherapy on GM of pediatric patients with hematologic malignancies. First, we confirmed that the GM profile of these patients is dysbiotic already at the onset of the disease [8,42], with low diversity and depletion of health-associated taxa, such as those belonging to the *Ruminococcaceae*, *Lachnospiraceae* and *Bifidobacteriaceae* families, compared to age/sex/BMI-matched healthy subjects. These GM features are typically associated with disorders of various kinds, potentially reflecting conditions of oxidative stress and reduced production of beneficial metabolites, such as short-chain fatty acids, with repercussions on the whole-body health [43]. As expected [44], this dysbiosis worsened during induction chemotherapy, with a further loss of diversity and especially an increase in pathobionts, namely enterobacteria and *Enterococcus*. In contrast, patients who received LF showed no such enrichment or variations in diversity, suggesting a potential stabilizing effect of LF on GM and possibly a reduced onset of complications related to therapy-induced dysbiosis. In support of this, we found a significantly lower incidence of febrile neutropenia, the most common complication of anticancer therapy [45], in the LF group compared to the placebo group. Febrile neutropenia has recently been associated with GM dysbiosis in pediatric allogeneic hematopoietic stem cell transplantation recipients, with longer fever duration being correlated with GM compositional instability and *Enterococcus* overabundance [17]. It should be remembered that patients with febrile neutropenia are at high risk of other complications, including BSI secondary to the translocation of pathogens from the gastrointestinal tract, especially in the context of mucosal damage induced by myeloblative chemotherapy regimens [46,47]. Although there were no differences in BSI episodes between groups, at least over the time frame of the study, the reduced proportions of pathobionts as observed in the LF group may be clinically relevant in reducing febrile neutropenia and infective complications. Enterobacteria and enterococci are, in fact, among the most common etiological agents of BSI [48,49], and their overabundance (especially of *Enterococcaceae*) at any time during chemotherapy in newly diagnosed pediatric ALL (i.e., the most prevalent disease in our study cohort) has been suggested to predict the later onset of infection [13]. Furthermore, high proportions of *Enterococcus* during ALL induction treatment were correlated with increased severity of intestinal mucositis, higher levels of C-reactive protein and decreased citrulline (a marker of functional enterocytes mass), thus contributing to exacerbate intestinal inflammation and compromise gut epithelial integrity [50,51]. It is, therefore, tempting to speculate that LF exerted an overall protective effect in the gut and beyond, counteracting the onset of dysbiotic signatures, particularly the increase in *Enterococcus*, and potentially favoring a reduction in therapy-related complications. No less relevant, supplementation with LF was associated with reduced amounts of *Akkermansia*, a genus that has recently been linked to the onset of febrile neutropenia in both adults and children undergoing intensive chemotherapy for acute leukemia, probably through erosion of the mucus barrier, resulting in bacterial translocation, endotoxemia, oxidative stress and an inflammatory/pyrogenic effect [17,52]. However, it should be remembered that *Akkermansia* has also been identified as a predictor of response in patients with other cancers who received immunotherapy with PD-1 blockade, possibly through Toll-like receptor-2-dependent enhancement of antitumor immune responses. The controversy over its beneficial or harmful role in the tumor context (and not only) will probably continue to persist, at least until mechanistic studies are available in the different pathological contexts in which *Akkermansia* appears clinically relevant. Furthermore, patients who received LF did not experience any reduction in *Oscillospira*, a health-associated taxon capable of producing short-chain fatty acids, such as butyrate, recently proposed as a next-generation probiotic candidate [53], while they showed enrichment in *Streptococcus* and *Veillonellaceae*. As for the latter, it should be mentioned that increased proportions of lactate utilizers from this family have recently been identified as distinctive signatures of therapeutic response in epithelial ovarian cancer, probably through interference with the lactate cycle and therefore with tumor progression [54]. On the other hand, it should be noted that patients who received placebo showed an increase in *Lactobacillaceae* over time. This family includes strains historically recognized as beneficial, but also others that have been shown to act as pathobionts in autoimmune-prone hosts, breaking through a dysfunctional gut barrier and promoting inflammatory responses [55] which emphasizes the need for high-resolution taxonomic studies along with mechanistic investigations.

The strength of our data is that they come from a randomized, double-blind, placebo-controlled trial. On the other hand, our study suffers from some limitations, including the small sample size, the applications of different induction chemotherapy regimens (although all considered to be at high risk of febrile neutropenia and gut mucosal damage) and the lack of long-term clinical outcomes. Moreover, it should be noted that the placebo and LF groups differed in basal weight. However, the baseline GM configuration was not different between the two groups, so it is plausible to assume that the weight difference did not affect GM dynamics. It should also be mentioned that, although oral LF was administered at a standard dose of 200 mg/die, not adjusted for body weight, all the main results were confirmed in a subgroup analysis on patients weighting less than 20 kg (and under 6 years of age). It is, therefore, possible to speculate that the administered dose exerted an effect on GM regardless of the patient’s weight, but further studies are needed to define the exact dose at which oral LF should be administered in order to maximize its benefits. Interestingly, no difference was observed in antibiotic use during induction chemotherapy between the two groups, thus further supporting the protective effect of LF.

In conclusion, we demonstrated that oral LF supplementation in pediatric patients affected by hematological malignancies receiving first-line induction chemotherapy was safe and counteracted therapy-related GM dysbiosis, by containing the overgrowth of pathobionts (e.g., *Enterococcus*) and modulating the abundance of other taxa potentially relevant to intestinal health (e.g., *Akkermansia*). Further studies in larger and more homogeneous cohorts are needed to validate these microbiological findings, which should employ other omics techniques (e.g., shotgun metagenomics, metatranscriptomics and metabolomics) and in vitro or animal models for mechanistic glimpses on the impact of LF on GM and host pathophysiology. The clinical benefit of oral LF supplementation, particularly in terms of reduction of infections and febrile neutropenia, will be determined in the ongoing AIEOP clinical trial.

## Figures and Tables

**Figure 1 pharmaceutics-14-01705-f001:**
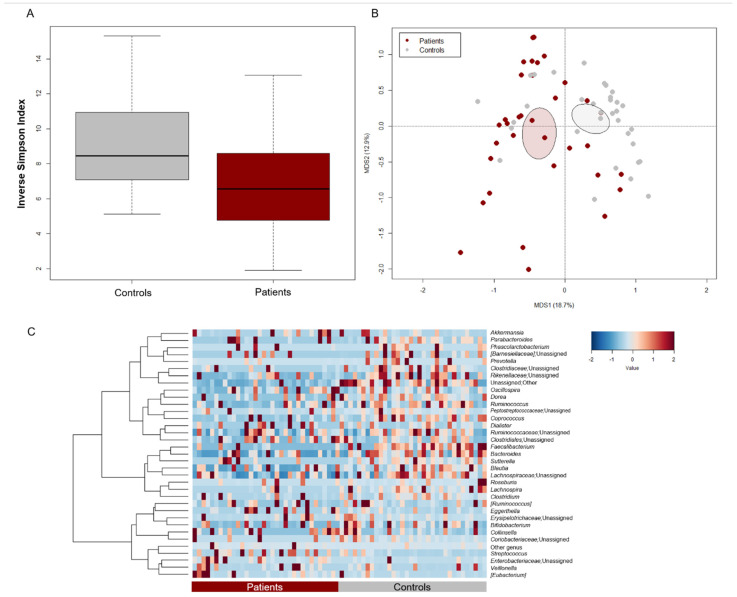
Gut microbiota structure of pediatric patients with hematological malignancies compared to healthy controls. (**A**) Boxplots showing the distribution of alpha diversity, estimated with the inverse Simpson index, in pediatric patients with hematological malignancies vs. age/sex/BMI-matched healthy controls from previous studies [38,39,40]. Patients were characterized by significantly less diversity (Wilcoxon test, *p* = 0.008). (**B**) PCoA based on Bray–Curtis dissimilarities between the gut microbiota profiles of patients and controls. Ellipses include 95% confidence area based on the standard error of the weighted average of sample coordinates. A significant separation between groups was found (permutation test with pseudo-F ratio, *p* = 0.001). (**C**) Heatmap showing Ward-linkage clustering based on Pearson’s correlation coefficients of the relative abundance of genera from patients and controls. Only taxa with relative abundance > 1% in at least 6 samples are shown.

**Figure 2 pharmaceutics-14-01705-f002:**
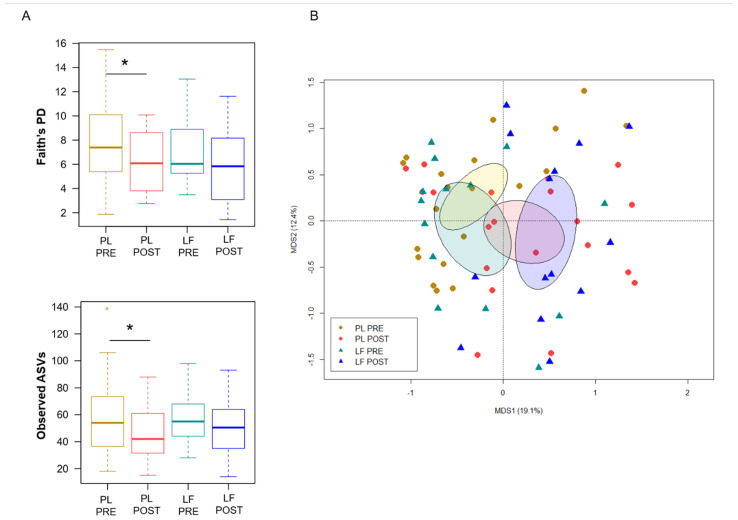
Gut microbiota diversity in pediatric patients with hematological malignancies receiving lactoferrin or placebo during induction chemotherapy. (**A**) Boxplots showing the distribution of alpha diversity, estimated with Faith’s phylogenetic diversity (Faith’s PD) and the number of observed ASVs, before (PRE) and after (POST) placebo (PL) or lactoferrin (LF) supplementation during induction chemotherapy (Wilcoxon test, * for *p* ≤ 0.05). (**B**) PCoA based on Bray–Curtis dissimilarities between the gut microbiota profiles of study groups. Ellipses include 95% confidence area based on the standard error of the weighted average of sample coordinates. A significant segregation among groups was found (permutation test with pseudo-F ratio, *p* = 0.005).

**Figure 3 pharmaceutics-14-01705-f003:**
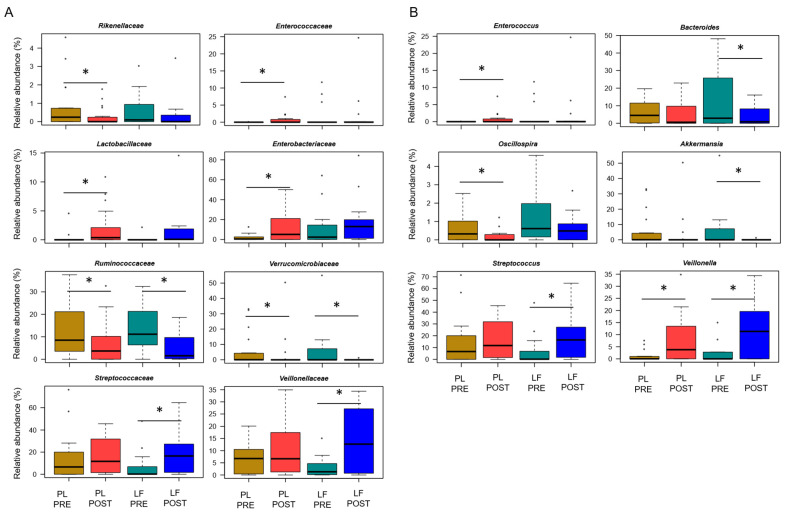
Gut microbiota changes in pediatric patients with hematological malignancies receiving lactoferrin or placebo supplementation during induction chemotherapy. Boxplots showing the relative abundance distribution of families (**A**) and genera (**B**) significantly differentially represented over time (PRE, before; POST, after), in relation to lactoferrin (LF) or placebo (PL) supplementation during induction chemotherapy (Wilcoxon test, * for *p* ≤ 0.05). Only taxa with relative abundance > 0.1% in at least 2 samples are shown.

**Table 1 pharmaceutics-14-01705-t001:** Summary of patient characteristics in the two groups.

	Lactoferrin (*n* = 14)	Placebo (*n* = 20)	*p*
Age at diagnosis: years, median (range)	3.50 (1.76–18.35)	6.96 (1.49–16.34)	0.42
Weight at diagnosis: kg, median (range)	15.0 (9.7–33.8)	24.0 (12.0–67.5)	0.03
Diagnosis, *n*, (%): ALL AML Lymphoma	13 (92.9) 1 (7.1) 0 (0.0)	17 (85.0) 1 (5.0) 2 (10.0)	0.47
Center, *n*, (%): Bologna Pavia Verona Perugia Bari Cagliari	4 (28.6) 0 (0.0) 1 (7.1) 2 (14.3) 7 (50.0) 0 (0.0)	6 (30.0) 1 (5.0) 1 (5.0) 2 (5.0) 8 (40.0) 2 (10.0)	0.78
Prior use of antibiotics, *n*, (%)	5 (38.5)	4 (20.0)	0.43
Neutropenic fever, *n*, (%)	8 (57.1)	18 (90.0)	0.04
Antibiotics during inductions (%)	10 (71,4)	14 (70.0)	1.00
BSI, *n*, (%)	6 (50.0)	7 (43.8)	0.74
Mucositis, *n*, (%)	5 (38.5)	6 (35.3)	0.90
Mucositis, grade: I II III IV	4 (80.0) 1 (20.0) 0 (0.0) 0 (0.0)	3 (50.0) 2 (33.3) 1 (16.7) 0 (0.0)	0.50

ALL: Acute Lymphoblastic Leukemia; AML: Acute Myeloid Leukemia; BSI: Blood Stream Infection.

## Data Availability

Sequence reads were deposited in the National Center for Biotechnology Information Sequence Read Archive (NCBI SRA; BioProject ID 856649).

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
