# Peer review of "Oral Lactoferrin Supplementation during Induction Chemotherapy Promotes Gut Microbiome Eubiosis in Pediatric Patients with Hematologic Malignancies"

_pharmaceutics, 2022, doi:10.3390/pharmaceutics14081705_

Round 1

Reviewer 1 Report

The study aimed to characterize the gut microbiome of children with hematologic malignancies by using 16S amplicon sequencing of the V3-V4 regions. The authors have clearly described the gaps in knowledge in the introduction by highlighting the lack of studies that have evaluated the use of lactoferrin to help maintain gut microbiome homeostasis in children with hematologic malignancies. These types of studies are indeed very interesting and needed, and demonstrate the importance of maintaining a healthy gut microbiome during treatments such as chemotherapy, which are known to have adverse effects on the microbiome.

Results are very interesting and optimistic, as they show that a degree of the gut microbiome diversity can be kept even after treatment with lactoferrin supplementation (Figure 2A). It is also intriguing to see that pre- and post- placebo groups are probably more similar amongst each other than pre- and post- lactoferrin supplementation group.

Have the authors looked at the clustering of the data in he PCoA plot based on subject?

The increase in the relative abundance of certain opportunistic pathogens in the placebo group after treatment confirms the gut microbiome dysbiosis that occurs after treatment. The increase in the relative abundance of certain bacterial taxa known to be beneficial is intriguing as well and consistent with previous results.

Do the authors have any explanation on the increase of the Lactobacillaceae in the placebo group? Wouldn’t you expect this group to increase with supplementation if it is overall considered a beneficial microbe?

In addition, while the explanation for the levels of Akkermansia seems reasonable, I’m concerned that other studies have found almost contradictory results in that Akkermansia seems to be a biomarker of treatment response and seems to be elevated in subjects that respond to treatment vs those that do not respond? Can the authors speculate on why this is the case? See reference below:

Derosa, L., Routy, B., Thomas, A. M., Iebba, V., Zalcman, G., Friard, S., ... & Besse, B. (2022). Intestinal Akkermansia muciniphila predicts clinical response to PD-1 blockade in patients with advanced non-small-cell lung cancer. Nature Medicine, 28(2), 315-324.

Author Response

The study aimed to characterize the gut microbiome of children with hematologic malignancies by using 16S amplicon sequencing of the V3-V4 regions. The authors have clearly described the gaps in knowledge in the introduction by highlighting the lack of studies that have evaluated the use of lactoferrin to help maintain gut microbiome homeostasis in children with hematologic malignancies. These types of studies are indeed very interesting and needed, and demonstrate the importance of maintaining a healthy gut microbiome during treatments such as chemotherapy, which are known to have adverse effects on the microbiome.

Results are very interesting and optimistic, as they show that a degree of the gut microbiome diversity can be kept even after treatment with lactoferrin supplementation (Figure 2A). It is also intriguing to see that pre- and post- placebo groups are probably more similar amongst each other than pre- and post- lactoferrin supplementation group.

We really thank the Reviewer for appreciating our work and for sharing the importance of preserving gut microbiome homeostasis during chemotherapy in order to reduce therapy-related adverse effects, as seen with lactoferrin supplementation.

Have the authors looked at the clustering of the data in the PCoA plot based on subject?

We thank the Reviewer for this valuable suggestion.

In the revised version of our manuscript, we have added a new supplementary figure, Figure S3, which shows the beta diversity of our entire dataset, where pre- and post-treatment samples from the same patient are linked to each other. Statistical analysis revealed significant segregation by subject (permutation test with pseudo-F ratio, p = 0.003). This finding was not unexpected given the well-known inter-individual variability of the gut microbiota, as also demonstrated in the specific context of pediatric acute lymphoblastic leukemia (the most prevalent disease in our study cohort) before, during and after chemotherapy (e.g., Chua et al., BMC Cancer. 2020 doi: 10.1186/s12885-020-6654-5). On the other hand, our data on post-treatment compositional variations suggest that lactoferrin is able to partly overcome this variability, inducing some shared changes of potential clinical relevance.

Please, see L243-245 of the Results section, where we briefly commented on the new data.

The increase in the relative abundance of certain opportunistic pathogens in the placebo group after treatment confirms the gut microbiome dysbiosis that occurs after treatment. The increase in the relative abundance of certain bacterial taxa known to be beneficial is intriguing as well and consistent with previous results.

Do the authors have any explanation on the increase of the Lactobacillaceae in the placebo group? Wouldn’t you expect this group to increase with supplementation if it is overall considered a beneficial microbe?

We thank the Reviewer for raising this point.

Indeed, lactobacilli are historically recognized as beneficial bacteria and many species are nowadays used as probiotics with proof of efficacy in different contexts. Nevertheless, there are some papers that report them augmented in pathological contexts and thus argue about their role in host physiology. This is the case, for example, of cardiometabolic (Silveira-Nunes et al., Front Pharmacol. 2020 doi: 10.3389/fphar.2020.00258) and immunological diseases (Wang et al., Front Immunol. 2022 doi: 10.3389/fimmu.2022.883747). In particular, some Lactobacillus strains have been shown to act as pathobionts in autoimmune-prone hosts, breaking through a dysfunctional gut barrier and promoting inflammatory responses (see for a review Fine et al., Adv Immunol. 2020 doi: 10.1016/bs.ai.2020.02.002).

In the revised version of our manuscript, we briefly discussed these contradictions, further emphasizing the need for high-resolution taxonomic studies along with mechanistic investigations (please, see L332-336).

In addition, while the explanation for the levels of Akkermansia seems reasonable, I’m concerned that other studies have found almost contradictory results in that Akkermansia seems to be a biomarker of treatment response and seems to be elevated in subjects that respond to treatment vs those that do not respond? Can the authors speculate on why this is the case? See reference below:

Derosa, L., Routy, B., Thomas, A. M., Iebba, V., Zalcman, G., Friard, S., ... & Besse, B. (2022). Intestinal Akkermansia muciniphila predicts clinical response to PD-1 blockade in patients with advanced non-small-cell lung cancer. Nature Medicine, 28(2), 315-324.

Once again, thank you for raising this point of considerable and current interest.

As with Lactobacillus, contradictory reports also exist for Akkermansia: for example, on the one hand, it is considered a marker of metabolic health and has been proposed (and tested in a proof-of-concept exploratory study – Depommier et al., Nat Med. 2019 doi: 10.1038/s41591-019-0495-2) as a next-generation probiotic candidate for obesity and related complications (see also the opinion delivered by the EFSA Panel on Nutrition, Novel Foods and Food Allergens (NDA), EFSA J. 2021 doi: 10.2903/j.efsa.2021.6780), on the other hand, as a mucus degrader, it could seriously affect intestinal permeability, worsening already compromised situations, as has been observed in the case of Parkinson’s disease (Nishiwaki et al., Mov Disord. 2020 doi: 10.1002/mds.28119). With specific regard to therapies, as correctly stated by the Reviewer, Akkermansia has been identified as a microbial predictor of response in the context of epithelial tumors (Routy et al., Science. 2018 doi: 10.1126/science.aan3706) and advanced non-small-cell lung cancer (Derosa et al., Nat Med. 2022 doi: 10.1038/s41591-021-01655-5) treated with immune checkpoint inhibitors (specifically anti-PD-1). The underlying mechanisms are only now being appreciated (involving interleukin-12 or Toll-like receptor-2), so it is difficult to speculate on why the microorganism behaves differently in different settings; probably the pathological context and the therapeutic regimen play a decisive role.

In the revised version of our manuscript (Discussion section), we briefly discussed these contradictions (L319-325, 357).

Reviewer 2 Report

In this work, the dysbiosis caused by chemotherapy in pediatric patients with oncological diseases, mainly acute lymphoblastic leukemia, was studied. The quality of the microbial flora is analyzed before and after two weeks of treatment with lactoferrin (LF). The work is well written and it is original and important. I would like the authors to respond to the following questions and comments:

1. Do you use apo or holoLF? Is it from human or bovine origin? Which was the iron content?
2. In the study, why were children who had multiresistant bacteria excluded?
3. Why was LF not given to a patient with lymphoma? Perhaps the authors should remove the patients
with lymphoma and with acute mieloid leukemia, since they are very few.
4. How was the appropriate dose of LF for the study determined?
5. Were any leukemia markers measured to see its progress during treatment with LF?
6. Is it recommended to continue the treatment with LF all the time?
Or only during chemotherapy?
7.
Explain why you have very wide standard deviations.

Author Response

In this work, the dysbiosis caused by chemotherapy in pediatric patients with oncological diseases, mainly acute lymphoblastic leukemia, was studied. The quality of the microbial flora is analyzed before and after two weeks of treatment with lactoferrin (LF). The work is well written and it is original and important. I would like the authors to respond to the following questions and comments:

  1. Do you use apo or holoLF? Is it from human or bovine origin? Which was the iron content?

We utilized bovine lactoferrin, as reported in the main text, which possesses 69% sequence homology and identical functions with human Lf (hLf )and is the most used in both in vitro and in vivo studies. Regarding iron content, the iron saturation of the apo-lactoferrin we used was about 10%. (DOIhttps://doi.org/10.1007/s10534-022-00409-1)

  1. In the study, why were children who had multiresistant bacteria excluded?

In the main clinical study, analyzing the incidence of gram negative resistant infections and gut colonization by MDR bacteria in the two groups were among the secondary end-points. Therefore, patients with prior colonization were excluded.

  1. Why was LF not given to a patient with lymphoma? Perhaps the authors should remove the patients with lymphoma and with acute mieloid leukemia, since they are very few.

We opted to include patients based on the risk of febrile neutropenia associated with the intensity of the induction regimen, and not on the underlying disease, because the primary endpoint of this sub-study was to elucidate the effect of oral lactoferrin supplementation on the gut microbiota during induction chemotherapy considered at high risk for febrile neutropenia. However, if the Reviewer deems it more appropriate to present data on the ALL patient cohort alone, we will be more than willing to redo all analyses on this subgroup of patients.

  1. How was the appropriate dose of LF for the study determined?

We would like to thank the reviewer for raising this crucial point. Studies on preterm infant administered LF both on a standard dosage (i.e. PMID: 19809023) or based on weight (i.e. PMID: 27234409). Therefore, there is no consensus for the right dosage of LF. In our study, we opted to administer the standard dose of 200 mg/day, already reported in other studies, to facilitate administration (200 mg/day corresponded to 1 capsule/day) in order to increase the compliance in this subset of patients already taking several drugs and often affected by oral mucositis. Furthermore, considering it a pilot study, we speculated that, administering the same dosage to all the patients, we could unmask a correlation between the dose administered and the patients’ weight that could be analyzed in a subsequent dose-finding study.

  1. Were any leukemia markers measured to see its progress during treatment with LF?

This would be a very interesting topic to address. Our study was not designed to analyze leukemia progression, so any data would be biased and we opted to not include them in the dataset.

  1. Is it recommended to continue the treatment with LF all the time? Or only during chemotherapy?

The reviewer raised a very important and interesting point. Unfortunately, the literature is very scarce on this topic and we can not make any estrapolation based on our limited data. Anyway, we can speculate that a more prolonged administration of bLF, after the end of chemotherapy, could help in restoring a healthy microbiome without the risk of producing side effects.

  1. Explain why you have very wide standard deviations.

As regards the gut microbiota, the large standard deviations are not unexpected, on the contrary they underline the well-known inter-individual variability of this ecosystem. In this regard, as requested by Reviewer #1, in the revised version of our manuscript, we have provided a new supplementary figure, Figure S3, which shows the beta diversity of our entire dataset, in which pre- and post-treatment samples from the same patient are linked together. Statistical analysis revealed significant segregation by subject (permutation test with pseudo-F ratio, p = 0.003). This finding is in line with the available literature, including that specific to pediatric acute lymphoblastic leukemia (the most prevalent disease in our study cohort) before, during and after chemotherapy (e.g., Chua et al., BMC Cancer. 2020 doi: 10.1186/s12885-020-6654-5), and does not detract from our work, on the contrary it suggests that supplementation with LF is able to overcome this variability, inducing shared variations in the compositional structure of the microbiota.
